# Estimation of the lost productivity to the GDP and the national cost of correcting visual impairment from refractive error in Kenya

**Shadrack Muma** [1]* , **Kovin Shunmugam Naidoo** [1,2] , **Rekha Hansraj** [1]

1 Department of Optometry, College of Health Sciences, University of KwaZulu-Natal, Durban, South Africa,
2 OneSight EssilorLuxottica Foundation, Paris, France

☯ These authors contributed equally to this work.
* mumashadrack275@gmail.com

**Data Availability Statement:** All relevant data are within the manuscript and its Supporting Information files.

## Abstract

### Background

In developing countries such as Kenya, minimal attention has been directed towards population based studies on uncorrected refractive error (URE). However, the absence of population based studies, warrants utilization of other avenues to showcase to the stakeholders in eye health the worth of addressing URE. Hence this study estimated the lost productivity to the Gross Domestic Product (GDP) as a result of URE and the national cost required to address visual impairment from URE in Kenya.

### Methods

The lost productivity to the GDP for the population aged 16–60 years was calculated. Thereafter the productivity loss of the caregivers of severe visual impaired individuals was computed as a product of the average annual productivity for each caregiver and a 5% productivity loss due to visual impairment. The productivity benefit of correcting refractive error was estimated based on the minimum wage for individuals aged between 16–60 years with URE. Estimation of the national cost of addressing URE was based on spectacle provision cost, cost of training functional clinical refractionists and the cost of establishing vision centres. A cost benefit analysis was undertaken based on the national cost estimates and a factor of 3.5 times.

### Results

The estimated lost productivity to the GDP due to URE in in Kenya is approximately US$ 671,455,575 -US$ 1,044,486,450 annually for population aged between 16–60 years. The productivity loss of caregivers for the severe visually impaired is approximately US$ 13,882,899 annually. Approximately US$ 246,750,000 is required to provide corrective devices, US$ 413,280- US$ 108,262,300 to train clinical refractionists and US$ 39,800,000 to establish vision centres. The productivity benefit of correcting visual impairment is approximately US$ 41,126,400 annually. Finally, a cost benefit analysis showed a return of

**Funding:** The author(s) received no specific funding for this work.

**Competing interests:** The authors have declared that no competing interests exist.

US$ 378,918,050 for human resources, US$ 863,625,000 for corrective devices and US$ 139,300,000 for establishment of vision centres.

## Conclusion

The magnitude of productivity loss due to URE in Kenya is significant warranting prioritization of refractive error services by the government and all stakeholders since any investment directed towards addressing URE has the potential to contribute a positive return.

## Introduction

Refractive error (RE) which is a type of vision problem that makes it hard to see distant and near objects clearly is the leading cause of visual impairment (VI) globally [1]. Visual impairment (VI) occurs when an eye condition such as RE affects the visual system and the vision functions [2]. The magnitude of VI varies from mild which is the presenting visual acuity worse than 6/12, moderate VI which is the presenting visual acuity worse than 6/18 and severe VI which the presenting visual acuity is worse than 6/60 [3]. According to the Global Burden of Disease 2020, 338.8 million people globally had moderate and severe visual impairment (MSVI) with a projection of 535 million by 2050 if no interventions are taken to reduce avoidable blindness [4]. As a result, all stakeholders in eye health should direct efforts towards addressing uncorrected refractive error (URE) which is the presenting visual acuity of less than 6/12 in the better eye with an improvement of at least two lines after refraction through a simple pair of spectacles and contact lenses for net economic gain [5–7]. Holistically, eye care professionals should be at the forefront in providing evidence on URE and showcase to the government in particular, why policy changes and actions should be undertaken to address URE. In Kenya it is estimated that approximately 3.5 million individuals suffers from RE warranting the need for actions to address this condition [8]. To achieve this in developing countries such as Kenya, eye care professionals in collaboration with other stakeholders such as social entrepreneurs, Ministry of Health and commercial entrepreneurs should support population based studies. The population based studies would showcase the actual burden and influence the policy makers to establish policies and recognize the importance of addressing the burden attributed to URE [5]. Evidence from research shows that URE impacts negatively on the quality of life [9], therefore standard operating procedures and central monitoring should be in place to showcase to the government the burden attributed to URE from population based studies so as to influence the allocation of resources to address this condition in Kenya. The financial impact of vision loss on the global economy has been a major concern over the past decade [10–15]. As a result, showcasing the financial impact attributed to URE is desirable to influence the policy makers on the need for an action. Therefore, this study intended to establish the lost productivity to the GDP as a result of URE and the national cost required to address visual impairment (VI) from URE in Kenya. It is anticipated that this estimate can act as a baseline and showcase to the government, eye care professionals and all stakeholders in eye health the worth of directing efforts towards addressing accessibility, availability and affordability of refractive error (RE) services to achieve the aims of 2030 IN SIGHT [16].

## Materials and methods

According to Eckert and colleagues [10], MSVI results in a range of reduced productivity with 30%-55% not working while 45%-70% have their earning reduced by 35%. In Kenya, the minimum wage (MW) which is the lowest remuneration that employers can legally pay their employee is approximately US$ 101.505 per month [17]. The annual MW for the 45%-70% of the individuals with URE who are working was computed. Thereafter, a factor of 0.35 was used to estimate the reduced earnings for the visually impaired individuals who are working [10]. Currently in Kenya, approximately 3.5 million people are estimated to suffer from RE [8]. The method proposed by Eckert and colleagues [10] in calculating the lost productivity as a result of MSVI was adopted. The productivity loss was computed as a product of (0.45 of the affected population * 0.35 of reduced earnings) and (0.7 of the affected population * 0.35 of reduced earnings) [10]. The population aged 16–60 years with VI was included for the estimations with the assumption that the population below 16 years and those above 60 years with URE yield no economic productivity to a country's GDP as proposed by Smith and colleagues [18].

The productivity benefit of addressing URE among VI individuals in Kenya of working age (16–60 years) was estimated. In Kenya, approximately 51% of the VI individuals constitute to the working age group [19] with an employment rate (ER) of 16% [20]. In the absence of VI, those treated would be employed at the same rate as the individuals with good vision [21]. In Kenya, the ER is estimated at 38.5% nationally for the population aged 16–60 years with a MW of US$ 101.505 monthly [22]. To scale the current ER for VI individuals to the national average of 38.5%, additional investments beyond the recurrent expenditure in eye health care to develop sustainable primary and secondary healthcare systems are desirable. An assumption was made that if individuals with VI benefits from spectacles and low vision devices, then the ER would potentially be scaled to the national average of 38.5% [20]. An assumption was also made that if the individuals with VI who have attained the working age are provided with corrective devices and employed thereafter then the productivity benefit would increase. The productivity benefit was computed in three perspectives. Firstly, the productivity benefit for the 51% of the VI individuals who have attained the employment age was computed as a product of the annual MW and the 51% of the population with VI in Kenya [22]. The unadjusted productivity benefit was adjusted using the labour force participation rate (LFPR) which is the estimate of the economy's active workforce [23]. In Kenya, the LFPR is approximately 0.739 [24]. Secondly, the current productivity benefit accrued from the 16% of the VI in Kenya was calculated. Finally, a productivity benefit was computed with an assumption that the ER which is the measure of the extent to which available labour resources are being utilized is scaled to the national average of 38.5% for VI [23].

The productivity loss of the caregivers for individuals with severe VI to the GDP was computed. The population aged between 16–60 years with normal vision was included with the assumption that they are productive and can take care of individuals with VI. The average income of the normal sighted caregivers aged between 16–60 years annually was calculated. Given that approximately 3.5 million Kenyans have VI [8], an assumption was made that the population with severe VI would require some sought of care in the absence of corrective devices. Out of the 3.5 million Kenyans with MSVI, it is estimated that approximately 7.5% have severe VI [25]. It was assumed that there is one caregiver for every individual with severe VI. The number of required caregivers was estimated based on the number of individuals with severe VI in Kenya. Based on a study by Fricke and colleagues [26], it was assumed that every caregiver for an individual with VI experiences a productivity loss of 5% on the total average yearly productivity. Hence the productivity loss of caregivers was computed as a product of the average annual productivity for each caregiver and the 5% productivity loss due to the care

provided. It was assumed that if severe VI is addressed, it is expected that caregivers will be able to increase their productivity through employment, education and other approaches.

The national cost calculation entailed summation of the cost required to establish vision centres (VCs), the cost required to scale functional clinical refractionists and the cost required to provide spectacles based on the estimated prevalence of URE in Kenya. An assumption was made that URE cannot be addressed effectively in the absence of adequate VCs where patients with RE can access RE services. The cost required to establish the VCs in Kenya was computed based on estimates from the International Agency for the Prevention of Blindness (IAPB) of US$ 50,000 [16] and the Ministry of Health of Kenya of US$ 100,000 [27] per vision centre. The number of required VCs was estimated based on the population density and the Ministry of Health in Kenya consideration in establishing a healthcare facility [27]. In Kenya, the Ministry of Health recommends establishment of health facilities in an area with a population density of 100,000 [27]. However, the services offered in such healthcare facilities may not be inclusive of RE services.

The national cost estimate required to scale human resources was undertaken. The calculations conducted were based on the Ministry of Health recommendations on the ratios of human resource to population in Kenya [27]. The recommendations by the Ministry of Health for eye care professionals to population includes: 1:250,000 ophthalmologists, 1:250,000 optometrists, 1:100,000 ophthalmic clinical officers, 1:100,000 optometry technologists, 1:125,000 ophthalmic clinical officers/cataract surgeons and 1: 100,000 ophthalmic nurses [28]. Even though the scope of practice for all eye care professionals in Kenya allows the above categories of eye care professionals to undertake refraction, not all meets the threshold of a functional clinical refractionist who undertakes refraction 100% of their time [29]. The World Health Organization (WHO) recommendations on the minimum eye care teams for one per million population includes 4 ophthalmologists, 10 optometrists, 200 primary healthcare workers and 10 allied health professionals [30]. The key activities proposed by the WHO recognize the potential of optometrists in meeting the threshold for a functional clinical refractionist [30]. Based on the range of services that ophthalmic workers in Kenya undertakes as per their scope of practice [29], this paper estimated that ophthalmologists may spend roughly 20% of their time doing refraction [31], ophthalmic clinical officers roughly 30% of their time, optometrists may spend approximately 90% of their time and finally ophthalmic nurses may spend roughly 15% of their time [29]. This estimation was intended to provide a justifiable ground for stakeholders in eye health to direct efforts towards scaling eye care professionals meeting the threshold for functional clinical refractionists in Kenya. The number of required functional clinical refractionists from the existing eye care professionals having the potential to conduct refraction in Kenya was calculated based on the estimates by Morjaria and colleagues [32]. The deficit of functional clinical refractionists was computed based on the WHO recommendation of one functional clinical refractionist per 100,000 populations [30]. Even though the WHO recommendation is ideal, this paper presents different ratios which could be achieved through integration of technological approaches such as telemedicine [33]. As a result, this study estimated the number of functional clinical refractionists that should be trained to undertake refraction if adequate resources are available and telemedicine integration is prioritized. The calculations were based on the ratio of one refractionist per 50,000, 25,000 and 10,000 population. An assumption was made that if one refractionist per 50,000, 25,000 and 10,000 population ratios are to be adopted strengthened telemedicine integration is desirable. The estimated cost for training the deficit of functional clinical refractionists was computed based on the existing each eye care professionals in Kenya who operates as functional clinical refractionists. The cost required to train functional clinical refractionists was computed

for each category of the eye care professional based on the deficit and the recommendations by the WHO, Ministry of Health and this study suggestion based on telemedicine integration.

Finally, based on the national cost required to provide the 3.5 million individuals with URE in Kenya with spectacles [8], a review of the current private sector rates, as well as those of social enterprises (SE) was undertaken with an aim of estimating the cost required to provide RE services for the population in dire need of these in Kenya. The public health sector in Kenya was excluded as they do not offer spectacles but instead refer patients to the private sector for spectacles. The SE located within the public sector such as the Kenya Society for the Blind were included. To estimate the number of individuals with URE who already have corrective devices, the 6% or 210,000 estimates of the number of spectacles wearers in Tanzania was adopted [34]. The estimated number of individual with URE already having corrective devices was excluded from the national cost estimates. The national cost for correcting URE for the 3.5 million Kenyans was computed based on the average spectacles charges from the private sector and the SE sectors.

After the national cost estimates for addressing URE in Kenya were provided, a cost benefit analysis was conducted to show whether it would be worth investing the total amount required to provide corrective devices, scaling human resources and establishing VCs in Kenya. Considering that the benefit of addressing URE remains significant and mostly overrides cost invested in addressing URE by a factor of 3.5 times [19], the implication is that if one dollar is invested in addressing URE would generate a return of US$ 3.56. Hence the cost benefit analysis was computed as a product of the cumulative national cost required to address URE and the factor of 3.5 times was used. The estimate was intended to showcase whether it is worth investing the national cost of addressing URE by stakeholders in eye health in Kenya.

This study was approved by Maseno University Ethics Review Committee in Kenya (reference MUERC/1051/22) and the Biomedical Research Ethics Committee in South Africa (reference BREC/00004105/2022). The ethics review committee waived the requirements for the informed consent. This waiver was attributed to the nature of data recorded which is available within the public domains with no human subject involved [35].

## Results

### Lost productivity of visually impaired individuals to the GDP

The annual MW for the 45%-70% of the working individuals with MSVI in Kenya is approximately US$ 1,918,444,500 –US$ 2,984,247,000. The estimated productivity loss to the GDP as a result of VI in Kenya is approximately US$ 671,455,575 -US$ 1,044,486,450 annually.

### Lost productivity by caregivers of visually impaired individuals to the GDP

The number of caregivers required to attend to the 7.5% or 262,500 Kenyans with severe VI is approximately 227,951 caregivers. The average income for a caregiver of the visually impaired person in Kenya is approximately US$ 101.505 monthly [17]. Hence the annual MW for the estimated 227,951 caregivers is approximately US$ 277,657,995. Therefore, if the 5% productivity loss as a result of taking care of the severe visually impaired individuals is adopted, then approximately US$ 60.90 per caregiver is lost annually. Hence cumulatively a total of US$ 13,882,899 is lost annually if the 7.5% of severe visually impaired individual receive care from the healthy population.

## National cost estimates for addressing uncorrected refractive error in Kenya

Based on the scope of practice for ophthalmic workers allowed to undertake refraction in Kenya [29], the estimates on the proportions of professionals who undertake refraction by Morjaria and colleagues [32] in Kenya and this study assumption is that ophthalmologists, ophthalmic clinical officers, optometrists and ophthalmic nurses may spend approximately 20%, 30%, 90% and 15% of their time respectively doing refraction [29]. This study estimated that approximately 560 functional clinical refractionists exist in Kenya. Based on this study suggestion of one functional clinical refractionist per 10,000 populations if telemedicine is integrated, 4,756 functional clinical refractionists would be required in Kenya as shown in Table 1.

The current spectacle charges from the private sector in Kenya is estimated at an average cost of US$ 75 [39] and that of SE an average of US$ 35 [40]. Hence, with the current rates, majority of the Kenyan population are unable to afford the spectacles as only 38.5% are employed [20]. In consideration of the estimated prevalence of URE in Kenya of 3.5 million [8], the estimated national cost for addressing URE using the private sector rate of US$ 75 is approximately US$ 262,500,00 while the estimated national cost based on SE rates of US$ 35 is approximately US$ 122,500,00. With the assumption that there is a population already having corrective devices, this study adopted the estimates from Tanzania in which 6% (210,000) of the population with URE are estimated to have spectacles already [34]. The population with RE who already have corrective devices was excluded from the estimates. Out of the 3.5 million individuals with RE in Kenya, approximately 210, 000 already have corrective devices. Therefore, the estimated national cost for addressing URE based on the private sector rate was approximately US$ 246,750,000 while the estimated national cost based on SE rates was approximately US$ 115,150,000.

## National cost of training human resource to address uncorrected refractive error in Kenya

Currently, the cost of training an eye care professional in Kenya is approximately US$ 1,680—US$ 20,450 [41]. Hence if human resource is to be scaled based on various ratios recommended by the WHO, the Ministry of Health and this study recommendations, then approximately 540 functional clinical refractionists will be required for 1:100,000, 1,080 functional clinical refractionists for 1:50,000, 2,160 functional clinical refractionists for 1:25,000 and 5,400 functional clinical refractionists for 1:10,000 ratio refractionist to population. The cost required to train the professionals based on the ratios above will be approximately US$ 413,280- US$ 108,262,300. Details are shown in Table 2.

Conventionally, if one VC is established to serve a population of 50,000, then approximately 1,080 functional VCs will be required to serve the Kenyan population conveniently given that currently the Kenyan population is approximately 54 million [44]. Using the International Agency for Prevention and Blindness (IAPB) recommendation of an average of US$50,000 for

**Table 1. Deficit of functional clinical refractionists required in Kenya based on different practitioner–population ratios.**

| Recommendations | Source | Required functional clinical refractionists | Existing functional clinical refractionists | Deficit |
|---|---|---|---|---|
| 1:250,000 | [36] | 190 | 560 | Exceeds |
| 1:100,000 | [29] | 475 | 560 | Exceeds |
| 1:50,000 | [37] | 951 | 560 | 391 |
| 1:25,000 | [38] | 1,942 | 560 | 1,382 |
| 1:10,000 | [28] | 4,756 | 560 | 4,196 |

**Table 2. Estimated cost of training human resource in eye health in Kenya based on professionals and proposed ratios.**

| Professionals | Cost (US$) of training to completion | Source | Existing functional clinical refractionists | Total cost (US$) for training functional clinical refractionist; n = required number | | | |
|---|---|---|---|---|---|---|---|
| | | | | Required (n = 540) 1:100,000 [29] | Required (n = 1,080) 1:50,000 [37] | Required (n = 2,160) 1:25,000 [38] | Required (n = 5,400) 1:10,000 [28] |
| Optometrist | 7,000 | [42] | 178 | 2,534,000 | 6,314,000 | 13,874,000 | 36,554,000 |
| Ophthalmologist | 20,450 | [41] | 106 | 8,875,300 | 19,918,300 | 42,004,300 | 108,262,300 |
| Ophthalmic Clinical Officer/CS | 3,004 | [43] | 164 | 1,129,504 | 2,751,664 | 5,995,984 | 15,728,944 |
| Optometry technologist | 1,680 | [43] | 294 | 413,280 | 1,320,480 | 3,134,880 | 8,578,080 |
| Ophthalmic nurse | 3,004 | [43] | 80 | 1,381 | 3,004,000 | 6,248,320 | 15,981,280 |

equipping and establishing a VC [45] and the Kenya Ministry of Health, [46] estimate of US $100,000, the estimated cost required to establish the VCs in Kenya are shown in Table 3.

## Productivity benefit to correcting the visually impaired individuals in Kenya

To scale the ER for visually impaired to the national average of 38.5%, additional 22.5% of the visually impaired should be employed. Out of the 3.5 million individuals in Kenya with VI [8], 1,785,000 have attained the employment age. Considering the MW of US$ 101.505 monthly [22], the unadjusted productivity benefit will be approximately US$ 2,174,237,100 annually if the 51% of the visually impaired individuals between 16–60 years are employed. This estimate was considered exceptional given that not all visually impaired individuals may meet the thresholds and requirements to be employed. As a result, the unadjusted productivity benefit was adjusted using the LFPR of 0.739. The adjusted productivity benefit is approximately US$ 1,606,761,216 annually. The second productivity benefit currently generated by the 16% of the visually impaired in Kenya is approximately US$ 41,126,400 annually. Finally, it was assumed that if the ER for visually impaired individuals is scaled to the national average of 38.5% then the productivity benefit will be approximately US$ 98,960,400 annually based on the current dollar exchange rates.

## Cost benefit analysis

The estimation of the cost benefit analysis was based on three aspects. The first aspect was the human resource. Based on this aspect, a cost benefit analysis was undertaken to show whether the cost of US$ 108,262,300 required in scaling the human resource would yield a positive return worth the investment. The number of the required human resources was determined based on the Ministry of Health recommendation of 1:100,000 and this study suggestion of 1:50,000, 1:25,000 and 1:10,000 for professional to the population. The second aspect was provision of corrective devices. This study computed a cost benefit analysis to show whether investing the cost of US$ 246,750,000 in providing corrective devices to the 3.5 million

**Table 3. Estimated cost of establishing the required VCs in Kenya.**

| VCs per population | Required VCs | Existing VCs | Deficit VCs | Cost of establishing the VCs | |
|---|---|---|---|---|---|
| | | | | IAPB Recommended rates (US$) | Ministry of Health, Kenya rates (US$) |
| 1:100,000 | 475 | 77 | 398 | 19,900,000 | 39,800,000 |
| 1:50,000 | 951 | 77 | 874 | 43,700,000 | 87,400,000 |

individuals with RE in Kenya was worth it. The final aspect was the establishment of VCs. This study determined the cost benefit analysis to show whether it is worth investing US$ 39,800,000 to establish VCs in Kenya. For each US$1 invested in the efforts to eliminate VI in Kenya, a return of US$3.56 could potentially be generated [19]. The cost benefit analysis estimate was computed as a product of the national cost estimates for scaling human resource, establishing VCs and providing corrective devices and the factor of 3.5 times. The result is shown in Table 4.

## Discussion

The growth of an economy's GDP could be determined by the population aged 16–60 years having the potential to undertake activities inclined towards enhancing net economic gain [12]. However, this can only be achieved if the population with disability is provided with the right devices to address their condition which limits them from undertaking their daily activities. Through a conservative approach, this study estimates a lost productivity of approximately US$ 671,455,575 -US$ 1,044,486,450 annually to the GDP as a result of URE. Similarly, for caregivers, the lost productivity to the GDP is estimated at approximately US$ 13,882,899 annually. This justifies the assertion that URE contributes to the country's GDP loss warranting the government to direct more efforts towards addressing URE. With the limited resources [28, 37, 48] during the COVID-19 pandemic, the government allocated resources to address the pandemic, a clear indication that evidence on the severity of a condition may force the government to take action and address the situation. Hence the burden of URE in Kenya should be showcased to the government by eye care professionals and other stakeholders to justify why attention should be directed towards addressing URE. However, there is lack of standard operating procedures and central monitoring as a minimum warranting the need for collaboration between the eye care professionals, researchers and all stakeholders in eye health. As it stands, the government and other stakeholders in eye health are directing efforts towards addressing URE based on global assumptions and estimates since a clear picture of the burden of URE in developing countries such as Kenya remains unknown. Therefore, the estimates provided in this paper are intended to act as a baseline to showcase to stakeholders in eye health the need for action to address URE.

Uncorrected refractive error contributes negatively to the GDP [49]. Dispensing a simple pair of spectacles not only impacts on the quality of life of an individual, but contributes to net economic gain [50]. This study shows that the productivity benefit of correcting VI in Kenya is approximately US$ 41,126,400 annually. This justifies the value of providing spectacles and other corrective devices to patients with RE. However, with approximately 59.6% to 85.7% of Kenyans living on less than US$ 1.9 a day [51], it would be necessary for the government to integrate spectacles and other corrective devices within the public sector so as to assist patients with URE to gain access and be able to afford the available RE services. More holistically, designing an approach where corrective devices are provided at prices based on the average income of everyone across the economic pyramid is desirable. This is to ensure that patients

**Table 4. Cost benefit analysis for investments towards addressing uncorrected refractive error in Kenya.**

| Area of investment | National cost estimate required (US$) | Factor estimate [47] | Return on investment (US$) |
|---|---|---|---|
| Human resources | 108,262,300 | 3.5 | 378,918,050 |
| Corrective devices | 246,750,000 | 3.5 | 863,625,000 |
| Vision centres | 39,800,000 | 3.5 | 139,300,000 |

with RE become productive and contribute to the GDP since any investment on URE contributes a return of a factor of 3.5 times [19].

Uncorrected RE remains the leading cause of VI globally [52] and Kenya is not an exception [8]. As a result, eye care professionals in collaboration with other stakeholders in eye health should be at the forefront in advocating for the inclusion of eye health within the public sector. Although the constitution of Kenya 2010 stipulates that the government should ensure access to quality healthcare [53], it is incapacitated with the private sector dominating the eye health sector. With the current charges for spectacles in the private sector of US$ 35 [39], the majority of the population across the economic pyramid cannot afford the spectacles as the poverty rate ranges between 59.6%-85.7% with the majority living on less than US$ 1.9 a day [51]. Therefore, achieving an effective RE coverage will require for the government partly recognizes the efforts that various stakeholders are making towards addressing URE. Tactically, considering the relatively reasonable charges by the SE, the government should establish policies to allow the SE, which are organizations that participate in business ventures through a commercial approach in order to fulfill a social purpose [54], to introduce RE services within the existing public sector institutions so as to scale effective RE coverage in a more cost effective and sustainable way.

Human resource development is a fundamental concept in a health system and capacity building strategies should be built on the community requirements [55]. While the human resources remains a major challenge within the eye care ecosystem in developing countries such as Kenya [37], a needs based approach of training professionals together with integrated telemedicine which is the mode of healthcare delivery through information and technology, is desirable [56, 57]. Telemedicine has been applied within remote settings to overcome geographical barriers to healthcare access, providing an alternative means of connecting patients to specialists [58]. A meeting abstract by Randhawa and colleagues [59] examining telehealth delivering subjective refraction found no statistically significant difference between in person and telehealth delivered subjective refraction. This study has shown that there are approximately 822 eye care providers who undertake refraction in Kenya. However, not all of them meet the threshold of functional clinical refractionists who undertake refraction one hundred percent of their time. As a result, out of the 822 eye care providers who undertake refraction, approximately 560 meet the threshold of functional clinical refractionists while others engage in various roles such as screening for eye diseases, eye surgery, training and treatment [29]. Therefore, if the recommendation of 1:50,000 is adopted as suggested by Fricke and colleagues [26] in Kenya, an additional 391 functional clinical refractionists are required. However, this study estimates that for effective RE coverage, a conventional recommendation of 1:10,000 is desirable and if adopted, 4840 clinical refractionists should be trained at a cost of approximately US$ 108,262,300. Therefore, considering the limited resources and the high cost of training for eye care professionals who end up not operating as functional clinical refractionists, needs based training should be used and focus on optometrists and integrated telemedicine to scale effective RE coverage in Kenya should be developed.

In Kenya, there are approximately 12,393 health facilities, with only 77 having functional optical units [32]. This low number of units in Kenya is linked to the lack of a demonstrated national prevalence of URE from population based studies which can showcase to the government the burden of URE and the need for action in this regard. For effective RE coverage to be achieved, the establishment of VCs is desirable. This study estimates that approximately 951 VCs are required for effective RE service delivery. This implies that one VC is desirable per 50,000. With limited resources in Kenya, a holistic approach of one VC per 100,000 is desirable warranting 475 of such units to be established in Kenya at a cost of US$ 39,800,000 based on the Ministry of Health of Kenya estimate of funding per VC of US$100,000 [27]. The

establishment of the VCs is costly; however, a cost benefit analysis still shows that a return will be accrued if such units are established. Therefore, this paper recommends integration of VCs within the existing healthcare facilities to offer RE services to a population of 100,000. This paper also recommends that in the presence of resources, one VC should serve a population of 50,000 with an assumption that this will enhance quality RE services delivery. Given the limited resources in Kenya to scale the VCs [32], This paper proposed a holistic conservative recommendation of one VC per 100,000 populations based on the Ministry of Health criteria for establishing healthcare facility [27]. Notwithstanding, this study also recommended that telemedicine integration into the eye health ecosystem should be adopted to ensure that the limited human resource available in Kenya could scale service delivery to the VCs. Therefore, the proposed telemedicine should entail a communication among eye care professionals in regards to a patient presentation and course of management and should be integrated within the established optical units to scale RE service delivery.

Addressing URE should conventionally be prioritized for quality life. Hence considering the current situation in which approximately 83% of VI is present in low-income and middle-income countries among the underserved population [60, 61], the productivity loss might be huge given the limited estimates showing the cost benefit analysis of URE. As a result, this paper will act as baseline for stakeholders in eye health especially the government to understand the value of investing the estimated national cost for addressing URE in Kenya. Addressing URE will not only be beneficial to the individuals but will also benefit the caregivers of individuals with URE and contribute to the economic growth of a country. This study has showed that any investment directed towards addressing URE contributes to a positive return. As a result, it is worthy of attention for stakeholders in eye health sector to invest more in addressing URE as a projected positive return of US$ 3.56 will be accrued provided the investment is properly executed by professionals.

In conclusion, URE contributes to lost productivity and reduction of the GDP in Kenya, therefore investing the national cost in training human resources, establishing VCs and providing spectacles to the population in dire need will contribute a productivity benefit. Although the estimates in this study may not be exact due to aspects like varying exchange rates for a dollar, to the best of our knowledge, this is the first study to be conducted in Kenya on the lost productivity to the GDP as a result of URE. It also provides an overview of the current situation in Kenya highlighting why investments towards addressing URE should be prioritized by the government and stakeholders in eye health.

## Supporting information

**S1 Data. Data for the study.**
(PDF)

## Acknowledgments

The authors would like to acknowledge the statistician Mr. Partson for assisting during data analysis.

## Author Contributions

**Conceptualization:** Shadrack Muma, Kovin Shunmugam Naidoo, Rekha Hansraj.

**Data curation:** Shadrack Muma.

**Formal analysis:** Shadrack Muma.

**Methodology:** Shadrack Muma.

**Supervision:** Kovin Shunmugam Naidoo, Rekha Hansraj.

**Validation:** Kovin Shunmugam Naidoo, Rekha Hansraj.

**Visualization:** Kovin Shunmugam Naidoo, Rekha Hansraj.

**Writing – original draft:** Shadrack Muma.

**Writing – review & editing:** Kovin Shunmugam Naidoo, Rekha Hansraj.

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
