## [Decision Letter · Decision Letter 0]

21 Dec 2023

PONE-D-23-29248Estimation of the lost productivity to the GDP and the national cost of correcting visual impairment from refractive error in KenyaPLOS ONE

Dear Dr. Muma,

Thank you for submitting your manuscript to PLOS ONE. After careful consideration, we feel that it has merit but does not fully meet PLOS ONE’s publication criteria as it currently stands. Therefore, we invite you to submit a revised version of the manuscript that addresses the points raised during the review process.

**ACADEMIC EDITOR: **

**Before acceptance please ensure the following has been addressed:**

Address the queries of each review as clearly as possible.Undertake extensive English Language edit of the entire manuscript

We look forward to receiving your revised manuscript.

Kind regards,

Godwin Ovenseri-Ogbomo, OD, MPH, PhD

Academic Editor

PLOS ONE

Journal Requirements:

2. In this instance it seems there may be acceptable restrictions in place that prevent the public sharing of your minimal data. However, in line with our goal of ensuring long-term data availability to all interested researchers, PLOS’ Data Policy states that authors cannot be the sole named individuals responsible for ensuring data access (http://journals.plos.org/plosone/s/data-availability#loc-acceptable-data-sharing-methods).

Additional Editor Comments:

Once again, thank you for submitting your manuscript to PLOS ONE. Three independent reviewers have suggested a number of corrections to be made to improve the manuscript. There is the additional concern about the presentation of the manuscript in terms of the English language. I therefore invite you to review the manuscript and submit a revised version. Also, rework the references to comply with the journal's guidelines/style.

Reviewers' comments:

Reviewer's Responses to Questions

**Comments to the Author**

1. Is the manuscript technically sound, and do the data support the conclusions?

Reviewer #1: Yes

Reviewer #2: No

Reviewer #3: Yes

2. Has the statistical analysis been performed appropriately and rigorously? 

Reviewer #1: Yes

Reviewer #2: No

Reviewer #3: Yes

3. Have the authors made all data underlying the findings in their manuscript fully available?

Reviewer #1: Yes

Reviewer #2: No

Reviewer #3: Yes

4. Is the manuscript presented in an intelligible fashion and written in standard English?

Reviewer #1: Yes

Reviewer #2: Yes

Reviewer #3: Yes

5. Review Comments to the Author

Reviewer #1: Thank you for your submission. The paper is relevant and highlights the importance of accessibility to vision services and optical corrective devices and the inequality that exists across the world. However, review of the paper with a recommendation of major review, is required.

Please see the main points below and detail comments in the manuscript:

As English is not your first language the paper would benefit form a proof-reading by a native speaker or a proof-reading service. Some sentences are formulated in an unclear fashion or are incomplete.

Try to avoid terms like cadres or refraction points. Alternatives have been provided in comments in the manuscript.

References should be reviewed and formatted in accordance to the publishers guidance. Some of them are incomplete, some have invalid links attached. Organisations such as WHO should be written in full in reference list.

In body of the manuscript, when referring to organisations, e.g. WHO and others, first mention name in full, then abbreviate.

Some of the estimations and calculations could be explained clearer.

The various ratios and recommendations should be referenced.

Think about how you would integrate telemedicine in your vision centres as in optometry, it has a limited role and can be used for triaging the patient, obtaining basic history 7 symptoms and for follow ups with the patient. This also would require a infrastructure (phones, internet for video calls with patient and other colleagues). If you are referring to telemedicine hubs and for eyecare professionals to communicate between themselves in regards to a patient/patient presentation or course of treatment or management, please specify that.

Reviewer #2: I find the topic interesting and important to the eye care services in Kenya. From my point of view, the paper needs to be revised and improved. First, the authors don´t provide a definition for Refractive error. That is a critical point since that will have an impact on the age group that should be considered to estimate productivity losses. It is unclear why they decided to use two age groups 16-50 and 16-60. Why should people above the age of 50 y.o be considered unproductive if the retirement age in the country is 60 y.o. They also don´t provide a definition for labour force participation and employment rate which I believe would be useful to do. Second, they use disability weights to measure productivity losses without providing a good rational to do it. Disability weights and productivity losses should not be seen as substitutes. They measure different aspects of life. Disability weights represent the magnitude of health loss while productivity losses represent “the production loss due to illness, disability and death of productive persons, both paid and unpaid”. Third, it is not clear why are they using the all population (either 16-50 or 16-60) and not just the population who have refractive error (which will vary with the prevalence rate of refractive error in Kenya). Forth, I think there is a mistake with the reference of the disability weights used in the paper. Salomon´s et al (ref 6) report disability weights of 0.003 for mild visual impairment, 0.031 for moderate visual impairment and 0.184 for severe visual impairment. The disability weights used in the paper are proposed by WHO. Authors should explain why they did not use the latest disability weights figures. Fifth, there is something wrong with table 1 and table 2 (and with the way productivity losses are estimated). The product between number of people (Column 1 population aged 16-50 years) and disability weights (Column 2 population aged 16-50 years) can not directly produce a value expressed in USD. Finally, the national cost estimation and cost benefit analysis are not well describe and needs to be revised. The reference list doesn´t follow common reference styles.

Reviewer #3: The study is interesting and makes significant contribution to the economics of uncorrected refractive errors in Kenya

The study will require language editing

Clarify how the average income of the normal sighted caregiver aged 16 – 60 years was calculated

Line 130 – 135. Why was the assumption of time spent in doing refraction leave out the cadre of optometrists?

Line 135 – 139. Are optometrists classified under the clinical refractionist cadre? Please explain to provide clarity.

Referencing style should follow journal’s guidelines.

6. PLOS authors have the option to publish the peer review history of their article (what does this mean?). If published, this will include your full peer review and any attached files.

Reviewer #1: No

Reviewer #2: No

Reviewer #3: No

---

## [Author Response · Author response to Decision Letter 0]

8 Jan 2024

Dear Dr. Ogbomo (Academic Editor – Plos One)

Detailed below are the revisions made to the article titled ‘Estimation of the lost productivity to the GDP and the national cost of correcting visual impairment from refractive error in Kenya (Manuscript. No. PONE-D-23-29248)’. The changes were made based on the comments/suggestions of Reviewer #1, Reviewer #2, Reviewer #3 and the Academic Editor. All changes have been tracked on the revised manuscript. The authors’ responses to queries received are formatted through word track changes.

ACADEMIC EDITOR: 

Before acceptance please ensure the following has been addressed:

This is noted

Address the queries of each review as clearly as possible.

This is noted and has been addressed across the document

Undertake extensive English Language edit of the entire manuscript

Editing has been undertaken across the document 

Reviewer #1: Thank you for your submission. The paper is relevant and highlights the importance of accessibility to vision services and optical corrective devices and the inequality that exists across the world. However, review of the paper with a recommendation of major review, is required.

Please see the main points below and detail comments in the manuscript:

As English is not your first language the paper would benefit form a proof-reading by a native speaker or a proof-reading service. Some sentences are formulated in an unclear fashion or are incomplete.

This is noted and has been addressed across the document as suggested in the reviewed manuscript 

Try to avoid terms like cadres or refraction points. Alternatives have been provided in comments in the manuscript.

This is noted. Changes have been made across the document as suggested

References should be reviewed and formatted in accordance to the publishers guidance. Some of them are incomplete, some have invalid links attached. Organisations such as WHO should be written in full in reference list.

This has been addressed as suggested

In body of the manuscript, when referring to organisations, e.g. WHO and others, first mention name in full, then abbreviate.

Thanks. This has been revised accordingly across the document

Some of the estimations and calculations could be explained clearer.

This is noted and has been addressed across the document. Line 80-264

The various ratios and recommendations should be referenced.

Yes the recommendations have been referenced. Line 153-156, 159-162, 237-238, 262-263

Think about how you would integrate telemedicine in your vision centres as in optometry, it has a limited role and can be used for triaging the patient, obtaining basic history 7 symptoms and for follow ups with the patient. This also would require a infrastructure (phones, internet for video calls with patient and other colleagues). If you are referring to telemedicine hubs and for eyecare professionals to communicate between themselves in regards to a patient/patient presentation or course of treatment or management, please specify that. 144-146

This is noted. “If you are referring to telemedicine hubs and for eyecare professionals to communicate between themselves in regards to a patient/patient presentation or course of treatment or management”. Yes this is what we referred to. Detailed on page 148-150

Reviewer #2: I find the topic interesting and important to the eye care services in Kenya. From my point of view, the paper needs to be revised and improved. 

This is noted and has been addressed as suggested

First, the authors don´t provide a definition for Refractive error. That is a critical point since that will have an impact on the age group that should be considered to estimate productivity losses. 

The definition has been provided in line 54-55

It is unclear why they decided to use two age groups 16-50 and 16-60. 

This is noted. We would like to clarify that the initial estimates was based on the first study on global estimate of the cost required to address URE and the lost productivity to the GDP. However, with the advancement in the approaches used, we have made the calculations using the latest development (A Simple Method for Estimating the Economic Cost of Productivity Loss Due to Blindness and Moderate to Severe Visual Impairment by Eckert and colleagues). Therefore, this study adopts the 16-60 years only. Details are provided in line 80-92.

Why should people above the age of 50 y.o be considered unproductive if the retirement age in the country is 60 y.o. 

The unproductive age in the study has been rectified as individuals below 16 years and those above 60 years. Line 91

They also don´t provide a definition for labour force participation and employment rate which I believe would be useful to do. 

This is noted. The definitions have been provided in line 108-113

Second, they use disability weights to measure productivity losses without providing a good rational to do it. Disability weights and productivity losses should not be seen as substitutes. They measure different aspects of life. Disability weights represent the magnitude of health loss while productivity losses represent “the production loss due to illness, disability and death of productive persons, both paid and unpaid”. 

This is noted. Based on the current estimate approach, the disability weights were not used instead the minimum wage, the affected population and reduced earnings was used as proposed in a study by Eckert and colleagues: A Simple Method for Estimating the Economic Cost of Productivity Loss Due to Blindness and Moderate to Severe Visual Impairment. Line 80-92 provides the details

Third, it is not clear why are they using the all population (either 16-50 or 16-60) and not just the population who have refractive error (which will vary with the prevalence rate of refractive error in Kenya). 

This is noted. The current estimate is based on the 3.5 million Kenyans with URE. Line 86-92 details 

Forth, I think there is a mistake with the reference of the disability weights used in the paper. Salomon´s et al (ref 6) report disability weights of 0.003 for mild visual impairment, 0.031 for moderate visual impairment and 0.184 for severe visual impairment. The disability weights used in the paper are proposed by WHO. Authors should explain why they did not use the latest disability weights figures. 

This is noted. The disability weight approach has been eliminated with adoption of the latest formula for estimating the productivity loss. Therefore there is shift from the initial formula to the latest. Details are provided in the methodology section line 80-211

Fifth, there is something wrong with table 1 and table 2 (and with the way productivity losses are estimated). The product between number of people (Column 1 population aged 16-50 years) and disability weights (Column 2 population aged 16-50 years) can not directly produce a value expressed in USD. 

We agree and based on the current formula, we have made changes in line 213-226

Finally, the national cost estimation and cost benefit analysis are not well describe and needs to be revised. 

This has been revised as proposed in line 287-305 and line 227-272 in the revised document

The reference list doesn´t follow common reference styles.

This is noted and has been addressed as instructed

Reviewer #3: The study is interesting and makes significant contribution to the economics of uncorrected refractive errors in Kenya

The study will require language editing

This is noted and we have edited across the document

Clarify how the average income of the normal sighted caregiver aged 16 – 60 years was calculated

The minimum wage in Kenya is approximately US$ 101.505. Given that approximately 7.5% of the individuals with URE in Kenya have severe VI, we computed the annual income with an assumption that they constitute the LFPR. Details are provided on line 219-226

Line 130 – 135. Why was the assumption of time spent in doing refraction leave out the cadre of optometrists?

Optometrists are core and in this study we consider them as functional clinical refractionists. The WHO outline that optometrists are well placed to address URE. Line 230-232 provides the estimates. 

Line 135 – 139. Are optometrists classified under the clinical refractionist cadre? Please explain to provide clarity. Yes, the details are in line 166-167

Referencing style should follow journal’s guidelines.

This is noted and has been addressed under references sub-heading

---

## [Decision Letter · Decision Letter 1]

8 Feb 2024

PONE-D-23-29248R1Estimation of the lost productivity to the GDP and the national cost of correcting visual impairment from refractive error in KenyaPLOS ONE

Dear Dr. Muma,

Thank you for submitting your manuscript to PLOS ONE. After careful consideration, we feel that it has merit but does not fully meet PLOS ONE’s publication criteria as it currently stands. Therefore, we invite you to submit a revised version of the manuscript that addresses the points raised during the review process.

**ACADEMIC EDITOR:**
**Please address the concern of the having the recommendation in the methods section of the manuscript. The recommendation should typically be in the discussion.**

We look forward to receiving your revised manuscript.

Kind regards,

Godwin Ovenseri-Ogbomo, OD, MPH, PhD, FAAO

Academic Editor

PLOS ONE

Journal Requirements:

Additional Editor Comments:

As note by Reviewer 1, the recommendations noted in the methods section should typically be made in the discussion section of the manuscript if the recommendations are from the current paper. Please address these and a few other comments suggested.

Reviewers' comments:

Reviewer's Responses to Questions

**Comments to the Author**

1. If the authors have adequately addressed your comments raised in a previous round of review and you feel that this manuscript is now acceptable for publication, you may indicate that here to bypass the “Comments to the Author” section, enter your conflict of interest statement in the “Confidential to Editor” section, and submit your "Accept" recommendation.

Reviewer #1: All comments have been addressed

Reviewer #3: All comments have been addressed

2. Is the manuscript technically sound, and do the data support the conclusions?

Reviewer #1: Yes

Reviewer #3: Yes

3. Has the statistical analysis been performed appropriately and rigorously? 

Reviewer #1: Yes

Reviewer #3: Yes

4. Have the authors made all data underlying the findings in their manuscript fully available?

Reviewer #1: Yes

Reviewer #3: Yes

5. Is the manuscript presented in an intelligible fashion and written in standard English?

Reviewer #1: Yes

Reviewer #3: No

6. Review Comments to the Author

Reviewer #1: The manuscript has been improved significantly - well one. There are still some mistakes in the use of language & some things like VI & URE or how to correct RE should be explained/defined. Also, look at methodology/results - study recommendations should be included in discussion not results or methodology. You can use suggestions or estimations for calculations (see comments in the text). Therefore, minor revisions are still needed.

Reviewer #3: The authors have addressed the all comments i raised satisfactorily. This has improved the quality of the paper

7. PLOS authors have the option to publish the peer review history of their article (what does this mean?). If published, this will include your full peer review and any attached files.

Reviewer #1: No

Reviewer #3: **Yes: **Samuel Kyei

---

## [Author Response · Author response to Decision Letter 1]

9 Feb 2024

Dear Dr. Ogbomo (Academic Editor – Plos One)

Detailed below are the revisions made to the article titled ‘Estimation of the lost productivity to the GDP and the national cost of correcting visual impairment from refractive error in Kenya (Manuscript. No. PONE-D-23-29248_R1)’. The changes were made based on the comments/suggestions of Reviewer #1, Reviewer #3 and the Academic Editor. All changes have been tracked In the revised manuscript. 

ACADEMIC EDITOR:

Please address the concern of the having the recommendation in the methods section of the manuscript. The recommendation should typically be in the discussion.

This is noted and has been transferred to the discussion as proposed. 

Additional Editor Comments:

As note by Reviewer 1, the recommendations noted in the methods section should typically be made in the discussion section of the manuscript if the recommendations are from the current paper. Please address these and a few other comments suggested.

This is noted and has been corrected in the revised version line 383-386. The other comments by reviewer 1 have been addressed across the document. 

Review Comments to the Author

Reviewer #1: The manuscript has been improved significantly - well one. There are still some mistakes in the use of language & some things like VI & URE or how to correct RE should be explained/defined

Thank you. The suggestion has been integrated in line 55-59 and 62-64 as suggested. 

Also, look at methodology/results - study recommendations should be included in discussion not results or methodology. You can use suggestions or estimations for calculations (see comments in the text). Therefore, minor revisions are still needed.

This is noted and such recommendations have been transferred to the discussion section line 383-386.

Reviewer #3: The authors have addressed the all comments i raised satisfactorily. This has improved the quality of the paper

Thank you

---

## [Editor Report · Decision Letter 2]

6 Mar 2024

Estimation of the lost productivity to the GDP and the national cost of correcting visual impairment from refractive error in Kenya

PONE-D-23-29248R2

Dear Dr. Muma,

We’re pleased to inform you that your manuscript has been judged scientifically suitable for publication and will be formally accepted for publication once it meets all outstanding technical requirements.

Kind regards,

Godwin Ovenseri-Ogbomo, OD, MPH, PhD, FAAO

Academic Editor

PLOS ONE
---

## [Editor Report · Acceptance letter]

8 Mar 2024

PONE-D-23-29248R2 

PLOS ONE

Dear Dr. Muma, 

I'm pleased to inform you that your manuscript has been deemed suitable for publication in PLOS ONE. Congratulations! Your manuscript is now being handed over to our production team.

Kind regards, 

on behalf of

Dr. Godwin Ovenseri-Ogbomo 

Academic Editor

PLOS ONE